# *Aestipascuomyces**dupliciliberans* gen. nov, sp. nov., the First Cultured Representative of the Uncultured SK4 Clade from Aoudad Sheep and Alpaca

**DOI:** 10.3390/microorganisms8111734

**Published:** 2020-11-05

**Authors:** Marcus Stabel, Radwa A. Hanafy, Tabea Schweitzer, Meike Greif, Habibu Aliyu, Veronika Flad, Diana Young, Michael Lebuhn, Mostafa S. Elshahed, Katrin Ochsenreither, Noha H. Youssef

**Affiliations:** 1Process Engineering in Life Sciences 2: Technical Biology, Karlsruhe Institute of Technology, 76131 Karlsruhe, Germany; marcus.stabel@kit.edu (M.S.); tabea.schweitzer@kit.edu (T.S.); meike-greif@web.de (M.G.); habibu.aliyu@kit.edu (H.A.); 2Department of Microbiology and Molecular Genetics, Oklahoma State University, Stillwater, OK 74074, USA; radwa@okstate.edu (R.A.H.); mostafa@okstate.edu (M.S.E.); 3Bavarian State Research Center for Agriculture, Central Department for Quality Assurance and Analytics, Micro- and Molecular Biology, Lange Point 6, 85354 Freising, Germany; Veronika.Flad@lfl.bayern.de (V.F.); Diana.Young@lfl.bayern.de (D.Y.); Michael.Lebuhn@lfl.bayern.de (M.L.)

**Keywords:** anaerobic gut fungi, sequence-guided isolation, *Neocallimastigomycota* SK4 lineage

## Abstract

We report on the isolation of the previously-uncultured *Neocallimastigomycota* SK4 lineage, by two independent research groups, from a wild aoudad sheep rumen sample (Texas, USA) and an alpaca fecal sample (Baden-Württemberg, Germany). Isolates from both locations showed near-identical morphological and microscopic features, forming medium-sized (2–5 mm) white filamentous colonies with a white center of sporangia, on agar roll tubes and a heavy biofilm in liquid media. Microscopic analysis revealed monocentric thalli, and spherical polyflagellated zoospores with 7–20 flagella. Zoospore release occurred through an apical pore as well as by sporangial wall rupturing, a duality that is unique amongst described anaerobic gut fungal strains. Isolates were capable of growing on a wide range of mono-, oligo-, and polysaccharide substrates as the sole carbon source. Phylogenetic assessment based on the D1–D2 28S large rRNA gene subunit (D1–D2 LSU) and internal transcribed spacer-1 (ITS-1) regions demonstrated high sequence identity (minimum identity of 99.07% and 96.96%, respectively) between all isolates; but low sequence identity (92.4% and 86.7%, respectively) to their closest cultured relatives. D1–D2 LSU phylogenetic trees grouped the isolates as a new monophyletic clade within the *Orpinomyces*–*Neocallimastix*–*Pecoramyces*–*Feramyces*–*Ghazallamyces* supragenus group. D1–D2 LSU and ITS-1 sequences recovered from the obtained isolates were either identical or displayed extremely high sequence similarity to sequences recovered from the same aoudad sheep sample on which isolation was conducted, as well as several sequences recovered from domestic sheep and few other herbivores. Interestingly, members of the SK4 clade seem to be encountered preferably in animals grazing on summer pasture. We hence propose accommodating these novel isolates in a new genus, *Aestipascuomyces* (derived from the Latin word for “summer pasture”), and a new species, *A. dupliciliberans*. The type strain is *Aestipascuomyces*
*dupliciliberans* strain R4.

## 1. Introduction

The herbivorous gut harbors a wide range of bacterial, archaeal, protozoan, and fungal communities that collectively mediate the transformation of plant biomass into fermentable sugars and short-chain fatty acids [1]. Within such complex assemblages, members of the anaerobic gut fungi (AGF, phylum *Neocallimastigomycota*) remain the most enigmatic group [2,3]. During the last few decades, understanding of AGF diversity, ecology, and metabolic capabilities has been accumulating, and it is now broadly agreed that AGF play an integral role in the anaerobic degradation of recalcitrant lignocellulosic material [4,5] through hyphal penetration of plant material and production of a wide array of polysaccharide-degrading enzymes [1,2,6]. 

To date, eighteen different cultured AGF genera have been described, the majority of which were isolated in the last few years [7,8,9,10,11,12,13,14,15,16,17,18,19]. Generally, AGF genera are distinguished morphologically based on several structural features such as thallus development pattern [7,16], zoospore flagellation, and thallus morphology. Monocentric thallus development (i.e., zoospore cyst germination is not accompanied by nucleus migration into the germ tube, resulting in anucleated rhizoidal system) is the most abundant among AGF genera, as opposed to polycentric thallus development (i.e., nucleus enters the germ tube, developing a nucleated rhizomycelium) that has thus far only been observed in three genera, *Anaeromyces* [18], *Orpinomyces* [7], and *Cyllamyces* [19]. Additionally, the majority of AGF genera produce monoflagellated zoospores (with 1–4 flagella), with only four genera thus far (*Orpinomyces* [7], *Neocallimastix* [16], *Feramyces* [14], and *Ghazallomyces* [15]) known to produce polyflagellated zoospores (7–30 flagella). Finally, all currently-described genera exhibit filamentous thalli, except for the phylogenetically-related genera *Cyllamyces* [19] and *Caecomyces* [8], both known to display bulbous thallus morphology. 

Culture-independent diversity surveys have clearly demonstrated that AGF diversity is much broader than previously inferred from culture-based approaches. Such studies have identified several novel, yet-uncultured lineages, mostly through the use of the ITS-1 and D1/D2 LSU regions as phylogenetic markers [20,21,22,23,24]. Despite multiple recent efforts to isolate and characterize novel AGF lineages [11,12,13,14,15], many candidate genera remain uncultured. A recent study combining amplicon-based diversity survey with isolation efforts suggested that the success of isolation of an AGF taxon is positively correlated to its relative abundance in a sample, and negatively correlated to the sample evenness [24]. Further, multiple culture-based [11,15], and culture-independent [20,24] studies have provided evidence that poorly-sampled animal hosts harbor a wide range of hitherto uncharacterized AGF taxa. Based on these observations, we adopted two strategies to isolate novel AGF taxa—a targeted sequence-guided isolation strategy, where samples harboring relatively-high proportions of yet-uncultured genera are prioritized for AGF isolation efforts, and a sampling strategy targeting animals from which no prior isolation efforts have been reported. Intriguingly, these efforts, driven by two different hypotheses, and sampling different animals (a wild aoudad sheep and a zoo-housed alpaca) from two different geographical locations (Texas, USA, and Baden-Württemberg, Germany) yielded almost identical strains of a hitherto-uncultured AGF lineage (SK4, originally identified in samples from New Zealand (NZ)). This study demonstrates the global distribution of AGF lineages across multiple continents and suggests that some yet-uncultured AGF genera are not refractive to isolation given the right sampling and isolation conditions. It also highlights the value of implementing a sequence-guided culturing approach as well as directing isolation efforts to poorly-sampled animals. 

## 2. Materials and Methods 

### 2.1. Samples

Fresh fecal and rumen contents were collected in sterile 50-mL falcon tubes from a wild aoudad sheep (*Ammoragus lervia*) during a hunting trip in Sutton County, Texas, USA in April 2018. The hunting parties had all appropriate licenses, and the animals were shot either on private land with the owner’s consent, or on public land during the hunting season. Fecal samples were collected from an alpaca (*Vicugna pacos*) at the Karlsruhe Zoo, Germany in April 2019. Tubes were filled completely to ensure the absence of oxygen. Aoudad sheep samples were stored on ice and transferred to the laboratory within 24 h, where they were either directly utilized for DNA extraction or stored at −20 °C. Alpaca fecal samples were stored at room temperature until the next day, on which they were used for isolation.

### 2.2. Isolation

The aoudad sheep sample exhibited a relatively-high abundance (76.6%) of the yet-uncultured SK4 lineage in a prior study [24], and hence was chosen for targeted enrichment and isolation. Isolation efforts were conducted on fecal, as well as rumen samples. Rumen samples used in the isolation process were stored unopened at −20 °C. Fecal samples were opened once in an anaerobic chamber (Coy laboratories, Grass Lake, MI, USA) to obtain 0.5 g for use in culture-independent diversity survey efforts, and then stored at −20 °C. Isolation efforts were conducted 22 months post-sample-collection and DNA extraction. Samples were enriched in autoclaved rumen fluid–cellobiose (RFC) medium [25] for 24 h at 39 °C. Enriched tubes were serially diluted into anaerobic rumen fluid medium (RF) supplemented with either 1% *w*/*v* cellulose or a (1:1) mixture of cellobiose and switchgrass (1% *w*/*v*), and an antibiotics mixture of 50 μg/mL kanamycin, 50 μg/mL penicillin, 20 μg/mL streptomycin, and 50 μg/mL chloramphenicol. Following enrichment, serial dilutions up to 10^−5^ were performed, and the dilution tubes were incubated for 3 days at 39 °C. Dilutions showing visible signs of growth (change in the color of cellulose, clumping and floating of the switch grass, or production of gas bubbles) were used to prepare roll tubes [26] using RFC medium with 2% agar. Roll tubes were incubated for 2–3 days at 39 °C, after which single colonies were transferred into RFC medium. Roll tube preparation and colony picking were repeated at least three times to ensure the purity of the obtained isolates. Obtained isolates were maintained via bi-weekly sub-culturing into RFC media. Cultures were stored on agar medium for long-term preservation as previously described [25].

Isolation efforts from the alpaca sample were conducted by suspending 1 g of fecal material in a 1:1 mixture of anoxic salt solutions A (g/l: KH_2_PO_4_ (3.0), (NH_4_)_2_SO_4_ (6.0), NaCl (6.0), MgSO_4_ ∙ 7H_2_0 (0.6), and CaCl_2_ ∙ 2H_2_O (0.6)] and B ([g/l: K_2_HPO_4_ (3.0)), followed by inoculation in serum bottles with enriched rumen fluid medium (ERF) adapted from the basal medium described in [12], and supplemented with dissolved xylan solution (0.2% *w*/*v*), cellobiose (0.2% *w*/*v*), and wheat straw milled to a particle size of 1 mm (0.5% *w*/*v*), and an antibiotics solution of 60 µg/mL each of penicillin sodium salt, ampicillin sodium salt, and streptomycin sulfate. Inoculated serum bottles were then incubated for 7 days at 39 °C in the dark. Fungal growth was monitored by light microscopy and serum bottles with signs of anaerobic fungal growth were then used to inoculate roll tubes followed by incubation for 4 days at 39 °C in the dark. Single colonies were transferred into fresh medium. Roll tube preparation and colony picking were repeated at least three times to ensure the purity of the obtained isolates.

### 2.3. Morphological Characterization

For aoudad sheep isolates, both light and scanning electron microscopy were utilized to observe various microscopic features at different growth stages. For light microscopy, fungal biomass was collected from an actively-growing 2–3-day-old culture in RFC medium. Fungal biomass was stained with lactophenol cotton blue for examination of various thallus features including hyphae, sporangia, zoospores, and other specific microscopic structures as previously described [13,14,15]. For nuclear localization, samples were stained with DNA-binding dye, 4, 6 diamidino-2-phenylindole (DAPI, final concentration of 10 μg/mL), followed by incubation in the dark for 10 min at room temperature. All light microscopy examinations were conducted using an Olympus^®^ BX51 microscope (Olympus, Center Valley, PA, USA) equipped with a Brightline DAPI high contrast filter set for DAPI fluorescence and a DP71 digital camera (Olympus, Center Valley, PA, USA). Sample preparation and fixation for scanning electron microscopy was conducted as previously described [13]. The prepared samples were then examined on a FEI Quanta 600 scanning electron microscope (FEI Technologies Inc., Hillsboro, OR, USA).

For alpaca isolates, light microscopy was performed using a Nikon^®^ Eclipse E200 with a DFK 23U274 camera (Imaging Source^®^, Bremen, Germany), while fluorescence microscopy (to visualize DAPI staining) was performed using a Zeiss^®^ Axio Imager Z1 at an excitation wavelength of 353 nm. Differential interference contrast microscopy (DIC) was used for generating image overlay.

### 2.4. Substrate Utilization

Growth of the type strain (R4) obtained from aoudad sheep was assessed by replacing the cellobiose in RFC medium with glucose, xylose, mannose, fructose, glucuronic acid, arabinose, ribose, galactose, sucrose, maltose, trehalose, lactose, cellulose, xylan, starch, inulin, raffinose, polygalacturonate, chitin, alginate, pectin, peptone, or tryptone at a final concentration of 0.5% *w*/*v* [13,14]. To assess substrate utilization in the alpaca isolate, the strain was grown in defined media adapted from [12] with omission of clarified rumen fluid and addition of trace metal (prepared according to [27]), vitamin solution (prepared according to [28]) and replacement of the cellobiose with 0.05% of hemicellulose, xylan, starch, crystalline cellulose, inulin, chitin, pectin, cellobiose, maltose, trehalose, lactose, sucrose, glucose, xylose, mannose, fructose, arabinose, ribose, galactose, glucuronic acid, or 0.5% of wheat straw. The ability of a strain to utilize a specific substrate was considered positive if it exhibited viable growth on the tested substrate after four successive transfer events [13,14,18]. All results were compared to substrate-free medium.

### 2.5. Phylogenetic Analysis and Ecological Distribution

For the aoudad sheep isolates, DNA was extracted from 10 mL of 2–3-day-old RFC-grown cultures of five strains using DNeasy PowerPlant Pro Kit (Qiagen Corp., Germantown, MD, USA) according to the manufacturer’s instructions. For the alpaca isolate, DNA was extracted from 1-week-old cultures grown in rumen-free medium supplemented with 0.5% *w*/*v* cellobiose using the Quick-DNA Fecal/Soil Microbe DNA Miniprep Kit (Zymo Research). The extracted DNA was used as a template to amplify the region encompassing ITS-1, 5.8S rRNA, ITS-2, and the D1/D2 domains of the 28S (LSU) rRNA gene using the primers ITS5F (5′-GGAAGTAAAAGTCGTAACAAGG-3′) and NL4R (5′-GGTCCGTGTTTCAAGACGG-3′) [14,15] with the following PCR protocols: for the aoudad sheep samples, initial denaturation at 94 °C for 5 min followed by 39 cycles of denaturation at 94 °C for 1 min, annealing at 55 °C for 1 min, and elongation at 72 °C for 2 min, and a final elongation step at 72 °C for 10 min and for the alpaca samples, initial denaturation at 98 °C for 30 s followed by 30 cycles of denaturation at 98 °C for 10 s, annealing at 62 °C for 30 s, and elongation at 72 °C for 90 s, and a final elongation step at 72 °C for 2 min. PCR amplicons were cloned into a TOPO-TA cloning vector (Life Technologies^®^, Carlsbad, CA, USA) or using a PCR Cloning Kit (NEB) following the manufacturers’ instructions, and were Sanger-sequenced at the Oklahoma State University DNA sequencing core facility (22 clones from five aoudad sheep strains), or Eurofins Genomics (14 clones from one alpaca strain). For every clone sequence obtained, the ITS-1, and the D1/D2-LSU regions were extracted in Mega7 [29] by trimming using the sequence of the ITS-1 reverse primer MNGM2 (CTGCGTTCTTCATCGTTGCG), and the sequence of the LSU forward primer NL1 (GCATATCAATAAGCGGAGGAAAAG), respectively. The trimmed sequences were aligned to anaerobic fungal reference ITS-1 and D1/D2-LSU sequences using MAFFT v7.471 [30], and the alignments were manually curated in BioEdit [31]. The refined alignments were used to construct maximum-likelihood trees to assess the phylogenetic position of the obtained sequences using IQ-TREE v2.0.3 [32]. The best model was selected using ModelFinder [33] and 1000 ultrafast bootstraps [34] were applied. *Gonopodya prolifera* was used as the outgroup (NR_132861 for ITS-1, JN874506 for LSU).

To assess the ecological distribution of this novel lineage, we queried the trimmed ITS-1 sequences against a manually curated *Neocallimastigomycota* ITS-1 database encompassing all known cultured genera, as well as yet-uncultured taxa previously identified in culture-independent studies [20,21,23,24,35,36] using BLASTN. Hits with significant sequence similarity (>87%) were evaluated by insertion into ITS-1 phylogenetic trees. We also queried the D1/D2 LSU dataset generated in our prior effort [24], and hits with >93% sequence similarity were further evaluated by insertion into D1/D2-LSU phylogenetic trees.

### 2.6. Data and Culture Accession

Clone sequences are deposited in GenBank under accession numbers MW019479-MW019500 for the aoudad sheep strains R1-R5, and MW049132-MW049145 for the alpaca strain A252.

## 3. Results

### 3.1. Obtained Isolates

Five rumen isolates (R1–R5) were obtained from a single wild aoudad sheep in Texas, USA. Concurrently one isolate, A252, was obtained from fecal samples of alpaca in Baden-Württemberg, Germany. Preliminary morphological and microscopic characterization as well as phylogenetic analysis (see Section 3.5) showed identical attributes for strains R1–R5 and only minimal differences between the R strains and strain A252. One isolate (strain R4) was chosen as the type strain for detailed characterization. Below, we present a detailed characterization of the putative novel genus, and we report on the morphology and phylogenetic affiliation of the isolates highlighting differences between strains R4 and A252 when appropriate.

### 3.2. Colony Morphology and Liquid Growth Pattern

On solid media, strain R4 formed circular, white filamentous colonies with a white center of sporangia (Figure 1a). Colony size ranged between 2 and 5 mm. In liquid media, strain R4 produced a heavy fungal biofilm-like growth that loosely attached to the tube’s glass surface (Figure 1b).

### 3.3. Microscopic Features

Zoospores: Strain R4 produced globose zoospores with an average diameter of 9.3 ± 2.1 μm (standard deviation for 60 zoospores, range: 5–14 μm) (Figure 2a). All zoospores were polyflagellated, with 7–20 flagella and an average flagellum length of 28.1 ± 4.8 μm (average ± standard deviation from 60 zoospores, range: 19–36 μm). Strain A252 produced slightly larger spherical zoospores (10–20 μm, average 14 μm) with slightly longer flagella (35–49 μm, average 42 μm) (Figure 3a).Thalli and sporangia: Zoospore germination in strain R4 resulted in monocentric thalli with highly-branched anucleated rhizoids (Figure 2b–e). Strain R4 displayed both endogenous and exogenous thallus development. Endogenous thalli developed as a result of enlargement of zoospore cysts into sporangia with one (Figure 2f), or two adjacent (Figure 2g) rhizoidal systems. Endogenous sporangia displayed different shapes and sizes including ovoid (20–70 μm length (L) × 15–45 μm width (W) (Figure 2f), rhomboid (30–70 μm L × 40–85 μm W) (Figure 2g), and elongated (25–90 μm L × 15–40 μm W) (Figure 2h). No intercalary or pseudo-intercalary sporangia (sporangia present between two main rhizoidal systems) were observed. Exogenous sporangia were mainly developed at the end of unbranched sporangiophores that ranged in length between 10 and 300 μm (Figure 2i,j). Wide flattened sporangiophores (Figure 2i) and sporangiophores ending with sub-sporangial swellings (Figure 2k) were also frequently encountered. Mature exogenous sporangia ranged in size between 40 and 90 μm (L) and 15 and 35 μm (W), and they exhibited different morphologies including obpyriform (Figure 2i), ellipsoid (Figure 2j), globose (Figure 2k), ovoid (Figure 2m), and constricted ellipsoid (Figure 2n). Sporangial necks (the point between sporangia and rhizoids) were either tightly constricted (Figure 2f,k,q), or broad (Figure 2j,l,n,o). The neck opening, port, was either narrow (Figure 2k) or wide (Figure 2l). On the other hand, only endogenous sporangia were observed for strain A252, with one (Figure 3b) or, less frequently, two adjacent (Figure 3c) rhizoidal systems. Endogenous sporangia were mainly globose (diameter up to 145 µm) (Figure 3d), and ellipsoidal (up to 76 µm L × 48 µm W) (Figure 3e).Zoospore release: Zoospore release in strain R4 was achieved through two mechanisms, either from an apical pore (Figure 2o), as previously observed in *Feramyces* [14]), or through rupturing of the sporangial wall (Figure 2p), as commonly observed in *Neocallimastix* [16]. To our knowledge, the simultaneous utilization of both mechanisms by a single strain has not been previously reported in other AGF taxa. Sporangial walls either stayed intact (Figure 2o) or completely disintegrated after zoospore discharge (Figure 2q).

### 3.4. Substrate Utilization

Strain R4 utilized a wide range of substrates as the sole carbon and energy source. These included monosaccharides, e.g., glucose, fructose, mannose, xylose, and glucuronic acid, but not arabinose, galactose, or ribose. Strain R4 was able to metabolize and vigorously grow on all disaccharides tested including cellobiose, lactose, maltose, sucrose, and trehalose. Among the polymers tested, strain R4 was able to grow on cellulose, xylan, starch, inulin, and raffinose, but not alginate, chitin, pectin, poly-galacturonate, peptone, or tryptone.

On the other hand, strain A252 grew on polysaccharides including wheat straw, hemicellulose, xylan, starch, and inulin, but did not grow on chitin, pectin, or crystalline cellulose. The disaccharides cellobiose, maltose, lactose, and sucrose supported the growth of strain A252, but trehalose was not utilized. Strain A252 was capable of utilizing the monosaccharides glucose, xylose, and fructose, but not mannose, arabinose, ribose, galactose, or glucuronic acid.

### 3.5. Phylogenetic Analysis and Ecological Distribution

Phylogenetic analysis using the D1/D2 domains of 28S rRNA gene (D1/D2 LSU) reproducibly grouped all six isolates in a single, monophyletic cluster within the *Orpinomyces*–*Neocallimastix*–*Pecoramyces*–*Feramyces*–*Ghazallamyces* supragenic clade (Figure 4a). The isolates obtained in the present study showed very low D1/D2 LSU inter-sequence length heterogeneity (749–751 bp long, average 750 bp), low within-strain divergence between copies (0–1.74%) as well as low inter-sequence divergence between strains (0–1.6%). The closest cultured representative to the obtained isolates was *Feramyces austinii* (MG584193; 92.4% similarity). When compared to amplicon sequences, D1/D2 LSU sequences of the isolates showed highest similarity (93–100%) to amplicon sequences assigned to the uncultured lineage SK4 [24], originating from fecal material of the same aoudad sheep individual whose rumen sample was used for the US isolation (n = 1338 sequences), as well as to sequences recovered from blackbuck deer (n = 4), elk (n = 2), domesticated horse (n = 1), miniature donkey (n = 2), mouflon ram (n = 2), and oryx (n = 2).

On the other hand, the obtained isolates showed a slightly-higher ITS-1 length heterogeneity (196–200 bp; average 197.5 bp), within-strain divergence between copies (0–4.38%) as well as inter-sequence divergence between strains (0–5.84%). ITS-1 phylogeny (Figure 4b) placed the obtained isolates again close to the genus *Feramyces*. A BLASTN search against our custom ITS-1 database identified 1327 sequences with ≥87% sequence similarity. All hits were affiliated with the SK4 clade (originally identified in domesticated sheep and red deer samples in New Zealand (NZ) [21,22]). The majority of hits were from the same wild aoudad sheep samples from which the US isolates were obtained (n = 1311), domesticated sheep (n = 5) previously reported in NZ [21], as well as oryx, blackbuck deer, horse, miniature donkey, mouflon, and elk (n = 11). Analysis of all available SK4-affiliated sequences obtained from prior studies [21,22,24], and the current study indicates a clade ITS-1 sequence divergence range of 0–13.2%, with two well-defined subclades. Interestingly, divergent ITS-1 sequences originating from one isolate routinely clustered within both clades (Figure 4b), precluding their allocation to two distinct species and highlighting the difficulty associated with species-level OTU assignments using only ITS-1 data in the *Neocallimastigomycota*.

Notably, it seems that members of the SK4 clade exhibit higher abundance when animals graze on summer pasture. For example, in New Zealand’s domesticated sheep, SK4 was only identified as part of the AGF community when the animals were grazing on summer but not winter pasture [21,22], suggesting a potential relationship between the enrichment of SK4 in the AGF community and the season feed type.

## 4. Discussion

Here, we report on the isolation and characterization of the previously uncultured *Neocallimastigomycota* lineage SK4 from the rumen contents of a wild aoudad sheep and feces of a zoo-housed alpaca. Phylogenetic analysis using the D1/D2-LSU region showed that the six isolates obtained formed a single, monophyletic cluster within the *Orpinomyces*–*Neocallimastix*–*Pecoramyces*–*Feramyces*–*Ghazallamyces* supragenus clade [24,37]. All hitherto-described members in this clade are characterized by the production of polyflagellated zoospores, with the notable and peculiar exception of the genus *Pecoramyces*, which produces monoflagellated zoospores. This suggests an acquisition pattern of zoospore polyflagellation at ~ 46.3 Mya (the most current estimate of this clade emergence per [37]), followed by a recent loss and reverting to zoospore monoflagellation for the relatively-recently-evolved genus *Pecoramyces* (current estimates of emergence at 19.1 Mya, [37]). Similarly, all members of this supragenus clade form monocentric thalli, with the exception of the genus *Orpinomyces* that is known to develop polycentric thalli, suggesting that the development of polycentric thalli is a recent independent event that happened multiple times in the *Neocallimastigomycota* (for example with the emergence of *Orpinomyces*, *Anaeromyces*, and *Cyllamyces*). The closest cultured representatives of the SK4 clade are the genera *Feramyces* and *Neocallimastix*. While the three genera share similar morphological and growth patterns (e.g., polyflagellated zoospores and monocentric thalli development), they exhibit several distinct macroscopic and microscopic features. For example, members of the SK4 genus produce zoospores with 7–20 flagella, as opposed to 7–16 for *Feramyces* [14], and 7–30 for *Neocallimastix* [16]. Additionally, SK4 members produce terminal sporangia, while the *Feramyces* genus members produce terminal, pseudo-intercalary, and sessile sporangia [14]. Also, and perhaps most notably, members of the SK4 genus show two zoospore release mechanisms; either through an apical pore or via rupturing of the sporangial wall. On the other hand, the majority of *Neocallimastix* genus members are known to release zoospores through complete rupturing and lysis of the sporangial wall (Figure 25 in [16], with only a few exceptions (e.g., [38,39]), while *Feramyces* members release zoospores through apical pores (Figure 2x in [14]). To our knowledge, the dual zoospore release mechanism has not been encountered before in any of the cultured AGF genera members and hence appears to be highly characteristic of the SK4 genus.

Within the microbial world, a large fraction of organisms remains uncultured. This is more commonly encountered within the bacterial and archaeal domains, although a similar pattern has been suggested for the Fungi [40,41,42,43]. Within the anaerobic fungal phylum Neocallimastigomycota, multiple putative novel genera were identified in culture-independent studies [20,21,35]. Failure to obtain these taxa in pure culture could be attributed to several reasons. First, some AGF taxa are extremely fastidious and might require special nutritional and culturing requirements, and hence would not grow using routinely-utilized isolation and enrichment protocols [1,44]. Second, some AGF taxa might exhibit a very limited ecological distribution pattern and could be confined to few phylogenetically-related animal hosts. Indeed, many recently isolated novel genera appear to be of limited distribution, being observed only in very few samples from which they have been successfully isolated (e.g., *Aklioshbomyces* from white-tailed deer, *Ghazallomyces* from axis deer, and *Khyollomyces* (AL1) in the *Equidae* [15]). We argue that, in addition to mere presence, the relative abundance of the target lineage in the sample could be an important determinant for isolation success in the AGF. Our recent efforts [24] suggest that while some AGF genera are generalists, present in low abundance in a large number of samples and are often readily recovered from these samples, e.g., *Orpinomyces*, and *Anaeromyces*, others show a clear correlation between the success of their isolation and their relative abundance within a sample, especially in samples where one or a few lineages make up the majority (>90%) of the AGF community.

While information on the AGF community in the alpaca sample that was used for isolation in Germany is currently lacking, we believe that the success of obtaining a cultured SK4 representative was largely dependent on its presence in high relative abundance in the samples used for isolation. Therefore, this study clearly demonstrates the value of the sequence-guided isolation strategy that was employed here, whereby samples are initially prescreened using culture-independent approaches followed by targeting promising samples exhibiting a high proportion of novel/wanted genera for isolation efforts using a wide range of substrates, sample types, and growth conditions. Evidently, this approach will unfortunately involve storing the samples at −20 °C for a certain amount of time to allow for sequencing and data analysis to be conducted. Nevertheless, while some AGF taxa might not survive prolonged freezing, we have been successful in recovering isolates from samples stored frozen, especially when tubes were unopened, or at least where repeated freezing and thawing cycles were avoided, and where tubes were filled to the top with little to no space for air [14]. Isolates obtained from aoudad sheep and alpaca were very similar morphologically and phylogenetically despite the differences in geographical locations (Texas, USA versus Baden-Württemberg, Germany) and handling procedures, samples used (rumen versus fecal samples), animal host phylogeny (*Bovidae* versus *Camelidae*), and gut type (ruminant versus pseudo-ruminant). This demonstrates the global distribution of AGF lineages across multiple continents, and suggests that some yet-uncultured AGF genera are not refractive to isolation, given the right sampling and isolation conditions.

Based on morphological, physiological, microscopic, and phylogenetic characteristics, we propose accommodating these new isolates into a new genus, for which the name *Aestipascuomyces* (from aestas, Latin for summer, and pastura, Latin for pasture, to indicate the apparent enrichment of the clade during animal feeding on summer pasture) is proposed. The type species is *Aestipascuomyces dupliciliberans* (to indicate the two zoospore release mechanisms exhibited by members of the clade), and the type strain is *Aestipascuomyces dupliciliberans* strain R4.

## 5. Taxonomy

*Aestipascuomyces* Marcus Stabel, Radwa Hanafy, Tabea Schweitzer, Meike Greif, Habibu Aliyu, Veronika Flad, Diana Young, Michael Lebuhn, Mostafa Elshahed, Katrin Ochsenreither, and Noha Youssef, gen. nov.

MycoBank ID: MB837524

Typification: *Aestipascuomyces dupliciliberans* Marcus Stabel, Radwa Hanafy, Tabea Schweitzer, Meike Greif, Habibu Aliyu, Veronika Flad, Diana Young, Michael Lebuhn, Mostafa Elshahed, Katrin Ochsenreither, amd Noha Youssef (holotype).

Etymology: *Aestipascuo* = derived from aestas, Latin for summer, and pastura, Latin for pasture; *myces* = the Greek name for fungus.

Obligate anaerobic fungus that produces globose polyflagellated zoospores (7–20 flagella). Zoospores germinate into determinate monocentric thalli with highly-branched, anucleated rhizoids that lack constriction and intercalary swellings. The clade is defined by the sequences MW019479– MW019500 and MW049132–MW049145 (ITS-1, 5.8S rDNA, ITS2, D1–D2 28S rDNA). The most genetically-similar genera are *Feramyces*, which is characterized by its polyflagellated zoospores (7–16) and monocentric thalli that usually produce a single terminal sporangium, and on some occasions produce pseudo-intercalary and sessile sporangia and *Neocallimastix*, which is characterized by the production of polyflagellated zoospores (7–30), monocentric thalli, and empty zoospore cysts that remain at the base of sporangiophores.

*Aestipascuomyces dupliciliberans* Marcus Stabel, Radwa Hanafy, Tabea Schweitzer, Meike Greif, Habibu Aliyu, Veronika Flad, Diana Young, Michael Lebuhn, Mostafa Elshahed, Katrin Ochsenreither, and Noha Youssef, sp. nov.

Mycobank ID: MB837526

Typification: The holotype shown in Figure 2b in this manuscript is derived from the following: U.S.A. TEXAS: Sutton county, 30.591 N and 100.138 W ~300 m above sea level, 3-day-old culture of strain R4, which is isolated from the frozen rumen content of a female aoudad sheep (*Ammotragus lervia*), collected in April 2018 by Mr. Jim Austin. Ex-type strain R4 is stored on solid agar media at 39 °C at Oklahoma State University, Department of Microbiology and Molecular Genetics.

Etymology: *duplicus* = Latin for double, *liberans* = Latin for liberating or releasing. The species epithet highlights the dual zoospore release mechanisms.

An obligate anaerobic fungus that produces globose (5–14 μm in diameter) zoospores with 7–20 flagella (19–36 μm long). Zoospores germinate into determinate monocentric thalli with highly branched anucleated rhizoids that lack constriction and intercalary swellings. Both endogenous and exogenous thalli developments are observed. Mature endogenous sporangia are mainly rhomboid (30–70 μm L x 40–85 μm W) and elongated (25–90 μm L x 15–40 μm W). Mature exogenous sporangia range in size between 40 and 90 μm (L) and 15 and 35 μm (W), and display a variety of shapes including obpyriform, ellipsoid, globose, constricted ellipsoid, and ovoid. Sporangiophores vary in length between 10 and 300 μm. Wide flattened sporangiophores and sporangiophores ending with sub-sporangial swellings are occasionally encountered. Zoospores are released either through an apical pore or through the lysis of the sporangial wall. Sporangial walls remain intact or are completely collapsed after zoospore release. The fungus produces white filamentous colonies with a white center of sporangia (2–5 mm diameter) on agar roll tubes and heavy fungal biofilm-like growth that does nott attach to the tube’s glass surface in liquid media. The strain is defined by the sequences MW019494- MW019497 (for ITS-1, 5.8S rDNA, ITS2, D1–D2 28S rDNA).

Additional specimens examined: Radwa Hanafy strains R1 (MW019480, MW019481, MW019483, MW019484, MW019485, MW019486), R2 (MW019482, MW019487), R3 (MW019488, MW019490, MW019491, MW019492, MW019494), and R5 (MW019498, MW019489, MW019499, MW019479, MW019500) (GenBank accession number of clones in parenthesis), isolated from the same frozen rumen content of a female aoudad sheep (*Ammotragus lervia*) from which the type strain was isolated, in April 2018. Marcus Stabel strain A252 (MW049132-MW049145) (GenBank accession number of clones in parenthesis), isolated from the feces of an alpaca (*Vicugna pacos*) from the Karlsruhe Zoo, Karlsruhe, Germany in April 2019.

## Figures and Tables

**Figure 1 microorganisms-08-01734-f001:**
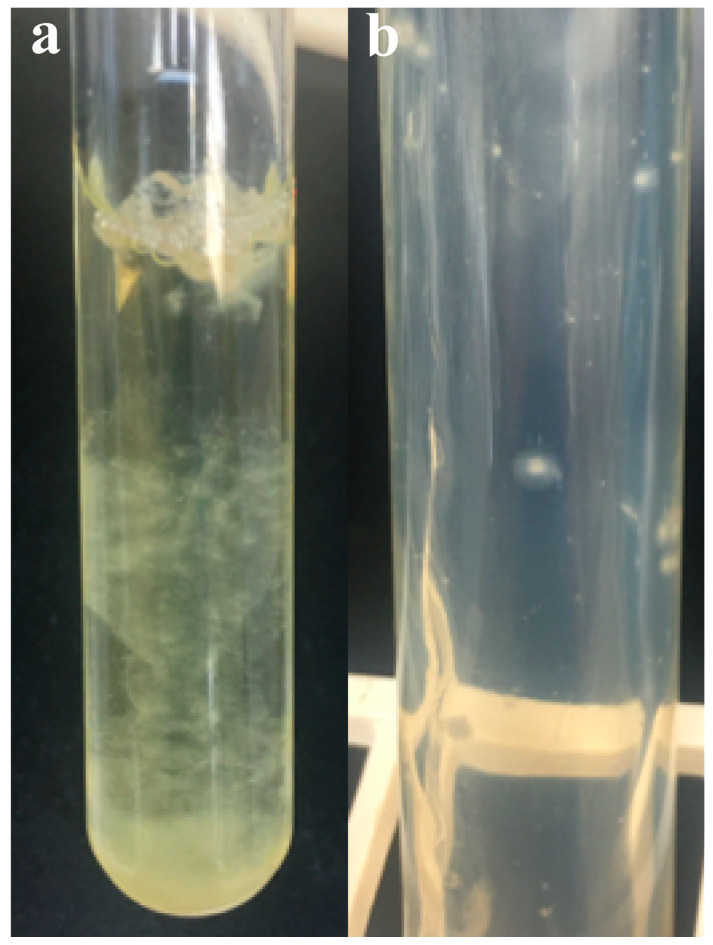
Macroscopic features of *Aestipascuomyces dupliciliberans* type strain R4. (**a**) Heavy fungal biofilm-like growth in liquid medium and (**b**) circular, white filamentous colonies with a white center of sporangia on cellobiose agar roll tube.

**Figure 2 microorganisms-08-01734-f002:**
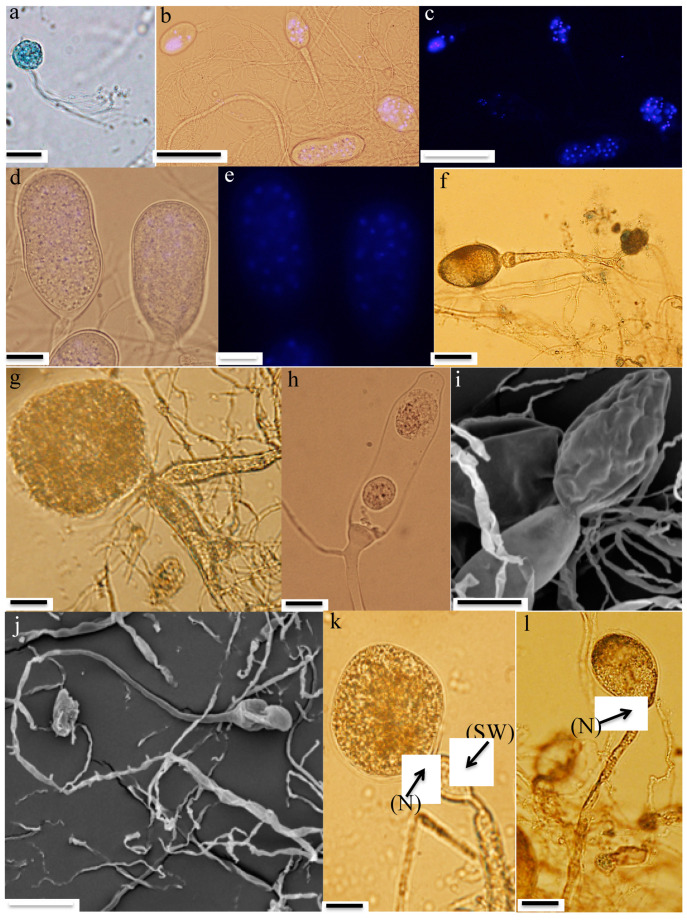
Microscopic features of *Aestipascuomyces dupliciliberans* type strain R4. Light (**a**–**h**, **k**–**n,** and **p**–**q**), fluorescence (**c** and **e**) and scanning electron (**i**, **j**, and **o**) micrographs are shown. (**b**,**c**) and (**d**,**e**) each depict the same field with (**c**) and (**e**) showing the fluorescence field, and (**b**) and (**d**) showing the overlay of fluorescence and phase contrast micrographs. (**a**) A spherical polyflagellated zoospore. (**b**–**e**) Monocentric thalli; nuclei were observed in sporangia, not in rhizoids or sporangiophore. (**f**–**h**) Endogenous sporangia: (**f**) ovoid sporangium with single rhizoidal system, (**g**) rhomboid sporangium with two adjacent rhizoidal systems, and (**h**) elongated sporangium. (**i**–**n**) Exogenous sporangia: (**i**) obpyriform sporangium on a flattened sporangiophore, (**j**) ellipsoidal sporangium on a long sporangiophore, (k) globose sporangium with sub-sporangial swelling and tightly-constricted neck, (**l**) ovoid sporangium with broad neck and wide port, (**m**) mature ovoid sporangium full of zoospores, and (**n**) constricted ellipsoidal sporangium. (**o**–**q**) Zoospore release mechanisms: (**o**) an empty sporangium with intact wall after zoospore release through an apical pore (arrow), (**p**) zoospore release through rupturing the sporangial wall, and (**q**) collapse and disintegration of the sporangial wall after zoospore release. (SW), sub-sporangial swelling; (N), neck. Bar = 20 μm (**a**, **f**–**h**, **k**–**n**, and **p**–**q**). Bar = 50 μm (**b**–**e**, **i**, and **o**). Bar = 100 μm (**j**).

**Figure 3 microorganisms-08-01734-f003:**
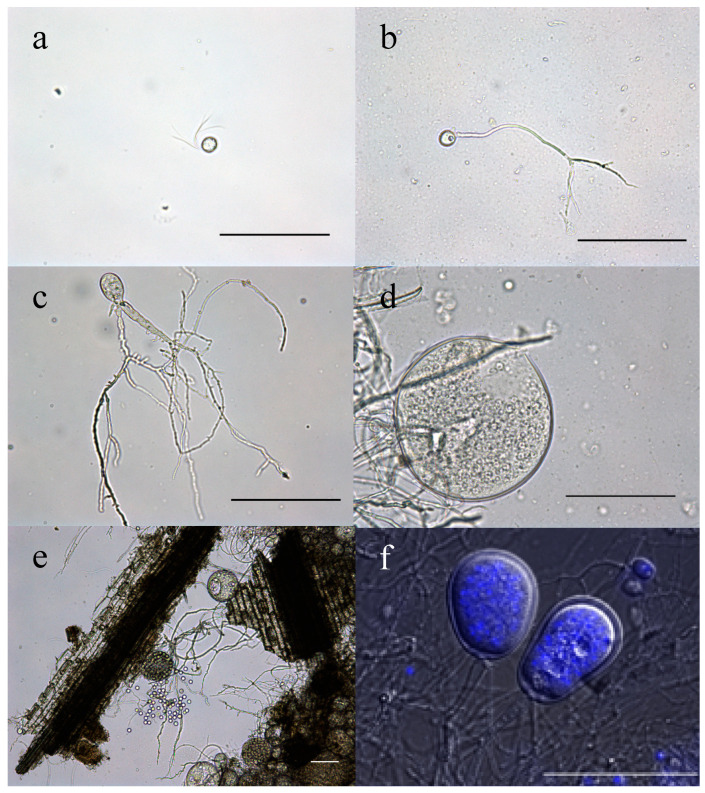
Microscopic features of *Aestipascuomyces dupliciliberans* type strain A252. Light (**a**–**e**) and DIC (**f**) micrographs. (**a**) A spherical polyflagellated zoospore with long flagella. (**b**–**e**) Endogenous sporangia: (**b**) young globose sporangium with a single rhizoidal system, (**c**) ovoid sporangium with two rhizoidal systems, (**d**) large globose sporangium with an apical pore for zoospore release, (**e**) sporangium during zoospore release, and (**f**) 4, 6 diamidino-2-phenylindole (DAPI)-stained mature ellipsoid sporangia. All scale bars are 100 μm.

**Figure 4 microorganisms-08-01734-f004:**
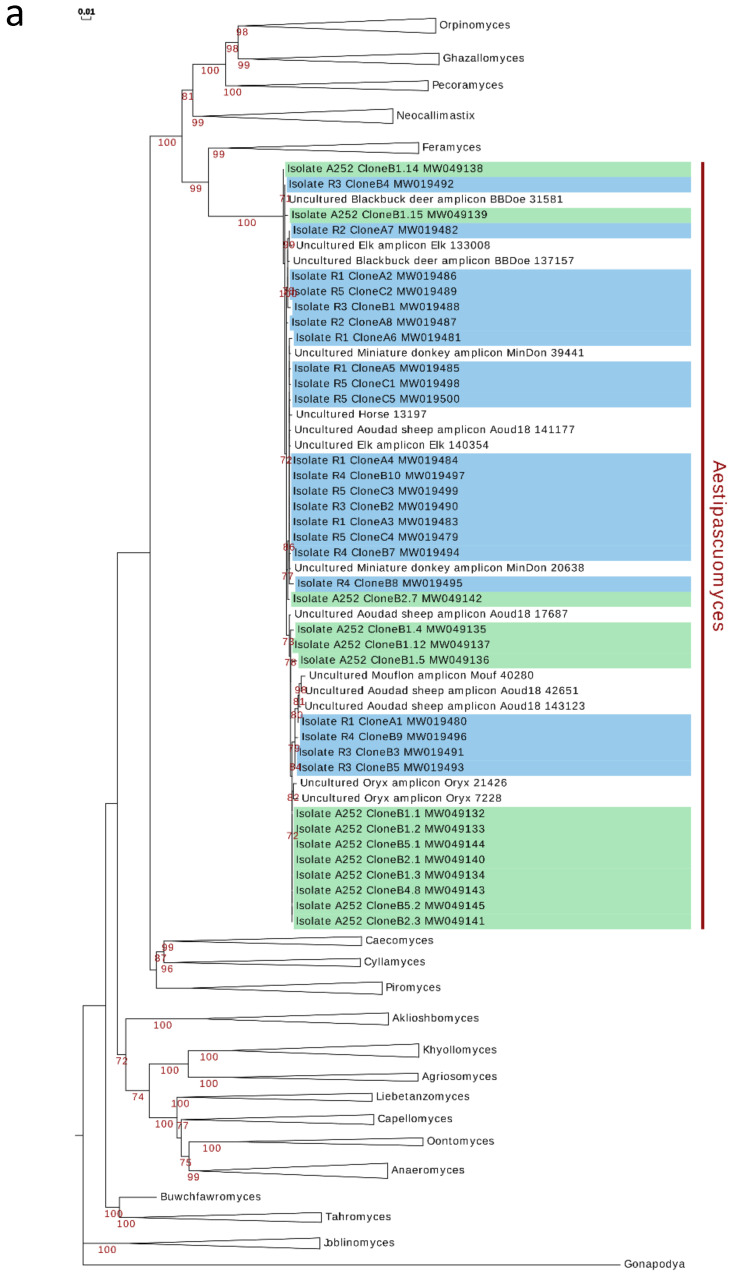
Phylogenetic affiliation of the *Aestipascuomyces* genus to other anaerobic gut fungi (AGF) genera based on the nucleotide sequences of the D1–D2 domains of 28S rRNA gene (**a**), and partial ITS-1 sequences (**b**). Sequences were aligned in MAFFT [30] and manually curated in BioEdit [31]. Curated alignments (LSU: 677 characters, 209 sequences; ITS: 295 characters, 126 sequences) were used to construct ML-trees using IQTREE with the predicted models TN+F+R2 (28S rDNA) or HKY+F+G4 (ITS) and –bb 1000. Bootstrap values are shown for nodes with more than 70% bootstrap support. Background color indicates the origin of the isolate (blue: Texas, USA; green: Baden-Württemberg, Germany).

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
