# Peer review of "Aestipascuomyces**dupliciliberans* gen. nov, sp. nov., the First Cultured Representative of the Uncultured SK4 Clade from Aoudad Sheep and Alpaca"

_microorganisms, 2020, doi:10.3390/microorganisms8111734_

Round 1
Reviewer 1 Report
Manuscript title: Aestipascuomyces dupliciliberatus gen. nov, sp. nov., the first cultured representative of the uncultured SK4 clade from Aoudad Sheep and Alpaca
Authors: Marcus Stabel et al., 2020
Journal name: Microorganisms- MDPI publisher
The manuscript is well written and, I appreciate the authors for the identified new species A. dupliciliberatus from the sheep rumen and fecal sample. Here, I give some suggestions that could enhance the quality of the content, particularly presentation of manuscript.
Minor comments:
- Author mentioned the rumen sample and fecal contents were collected from aoudad sheep and a zoo-housed alpaca respectively. Do you think how similarities of rumen between these two different organisms (in terms of function, structure and mechanisms)? Please clarify!
- Page 1: Line 25: Author mentioned rRNA subunit. Can you mention which subunit (16s, 28s, ..?)
- Manuscript title may add some catchy words about highlight of the study/work
- Page 3: Line 111-112: Chemical formulae should be subscript. Also, check carefully throughout manuscript.
- Line 132: Please provide the author name (reference)
- Materials and methods: provide sub-title numbers (2.1, 2.2…)
- Microscopic images missing scale bars.
Author Response
Reviewer 1.
The manuscript is well written and, I appreciate the authors for the identified new species A. dupliciliberatus from the sheep rumen and fecal sample. Here, I give some suggestions that could enhance the quality of the content, particularly presentation of manuscript.
Thank you so much for your kind words.
Minor comments:
- Author mentioned the rumen sample and fecal contents were collected from aoudad sheep and a zoo-housed alpaca respectively. Do you think how similarities of rumen between these two different organisms (in terms of function, structure and mechanisms)? Please clarify!
Thank you for your comment. We have added a few sentences on the differences in gut type between sheep and alpaca (families Bovidae and Camelidae) in the discussion. Now lines 403-409 in the revised manuscript.
- Page 1: Line 25: Author mentioned rRNA subunit. Can you mention which subunit (16s, 28s, ..?)
The abstract mentioned Large ribosomal subunit. We added 28S now for further clarification. (line 23 in revised manuscript)
- Manuscript title may add some catchy words about highlight of the study/work
Thank you so much for your suggestion. The main purpose of the manuscript is to report on the isolation and characterization of a new genus and the representative of a previously uncultured lineage. We feel that the current title succinctly delivers this message.
- Page 3: Line 111-112: Chemical formulae should be subscript. Also, check carefully throughout manuscript.
Thank you for pointing this out. We have corrected these throughout the manuscript.
- Line 132: Please provide the author name (reference)
We were not clear on what the reviewer meant here. The reference (#13) has already been cited under References.
- Materials and methods: provide sub-title numbers (2.1, 2.2…)
Done. Thank you.
- Microscopic images missing scale bars.
We apologize that the scale bars were hard to see on some of the figures. We changed the scale bars to white color on some figures to provide a better contrast with the dark background.
Reviewer 2 Report
Dear Authors,
in my opinion your work is very interesting in a cognitive context and contributes a lot to mycology and evolutionary taxonomy. Authors report on the isolation and characterization of the previously uncultured Neocallimastigomycota lineage SK4 from the rumen contents of a wild aoudad sheep and feces of alpaca. Based on morphological, physiological, microscopic, and phylogenetic characteristics, Authors proposed accommodating these new isolates into a new genus, for which is proposed name Aestipascuomyces with type species Aestipascuomyces dupliciliberatus.
All the figures are appropriate for this type of article. In general, the paper has a logical flow and fit the aims and scope of the journal. The abstract well correspond with the main aspects of the work. Nevertheless, I see a few weak points in this work (given below), which I am convinced that the Authors are able to resolve.
First of all, in the substantive context (just like the Authors themselves note), I see one weak point of this study, namely related to storage of the samples at -20oC. As Authors mentioned (quote) ,, while some AGF taxa might not survive prolonged freezing” it cannot be ruled out that avoiding this procedure the spectrum of cultivable fungi would be richer. The most important thing is that the Authors see this small technical problem and will certainly find the most optimal solution for the future.
As a reviewer I am also obligated to pay attention to the other less important weak points of this work and all mentioned below comments should be carefully considered.
Line 23
,,AGF strains” - all the abbreviations used within the manuscript should be expanded on first use, even those present in the abstract.
Line 24
,, on media containing a wide range of mono-, oligo-, and polysaccharides substrates.” sounds better.
Line 25
,,internal transcribed spacer-1 (ITS-1)” – as I know should be ,,internal transcribed spacer (ITS)”and the same for line 30 ,,ITS” not ,,ITS-1”. Please check throughout the manuscript
Line 31
In my opinion should be ,, sequences recovered from strain isolated from Aoudad sheep” and line 32 ,, as well as several sequences recovered from strains isolated from domestic sheep and few other herbivores”
Line 41
,,that collectively mediate in the transformation” sounds better
Line 43
,,Neocallimstigomycota” there is missing letter in this word
Line 90
,,until the next day on which they were used for isolation” is more adequate
Line 108
I think, there should be ,,obtained isolates”
Line 109, 132
,,as previously described in” – without ,,in”
Line 111-112
Subscripts should be used within molecular formulas of chemical compounds
Line 103 and 115
Please standardize the notation for units, for example Celsius degrees, for comparison: line 103, 115 and 117
Line 121
Should be ,,microscopes”
Line 123, 152
,,an actively growing 2-3 days old culture” sounds better
Line 142
There is ,, polygalcturonate” but should be ,,polygalacturonate” also check the line 239
Line 158 (the same in line 185)
There is unnecessary space: 5’- GGAAGTAAAAGTCGTAACAAGG-3` comparing to line 159
Line 219
There is unnecessary space ,,(Figure 2 n)”
Line 273
,,NZ” - all the abbreviations used within the manuscript should be expanded on first use, in this case (I think) the proper place to do this is line 77
Line 290
The is ,,phase contract micrographs” but should be ,,phase contrast micrographs”
Line 310, page 10, Figure 4
The name (caption) of clade below Tahromyces is unreadable. The same situation on page 11 below Orpinomyces
Line 313
I think it is worth to add ,,nucleotide sequences”
Line 373
Please, standardize the notation for units (Celsius degrees)
Line 377
,,tubes were filled to the top with little to no space for air” sound better for me
Yours sincerely,
Reviewer
Author Response
Reviewer 2:
Dear Authors,
in my opinion your work is very interesting in a cognitive context and contributes a lot to mycology and evolutionary taxonomy. Authors report on the isolation and characterization of the previously uncultured Neocallimastigomycota lineage SK4 from the rumen contents of a wild aoudad sheep and feces of alpaca. Based on morphological, physiological, microscopic, and phylogenetic characteristics, Authors proposed accommodating these new isolates into a new genus, for which is proposed name Aestipascuomyces with type species Aestipascuomyces dupliciliberatus.
All the figures are appropriate for this type of article. In general, the paper has a logical flow and fit the aims and scope of the journal. The abstract well correspond with the main aspects of the work. Nevertheless, I see a few weak points in this work (given below), which I am convinced that the Authors are able to resolve.
Thank you so much for your kind words.
First of all, in the substantive context (just like the Authors themselves note), I see one weak point of this study, namely related to storage of the samples at -20oC. As Authors mentioned (quote) ,, while some AGF taxa might not survive prolonged freezing” it cannot be ruled out that avoiding this procedure the spectrum of cultivable fungi would be richer. The most important thing is that the Authors see this small technical problem and will certainly find the most optimal solution for the future.
As a reviewer I am also obligated to pay attention to the other less important weak points of this work and all mentioned below comments should be carefully considered.
Line 23
,,AGF strains” - all the abbreviations used within the manuscript should be expanded on first use, even those present in the abstract.
Spelled AGF out in the abstract. (line 21 in revised manuscript)
Line 24
,, on media containing a wide range of mono-, oligo-, and polysaccharides substrates.” sounds better.
To avoid confusion this could bring about the presence of more than one carbon source in the growth media, we added as sole carbon sources to the end of this sentence in the abstract. (lines 22-23 in revised manuscript)
Line 25
,,internal transcribed spacer-1 (ITS-1)” – as I know should be ,,internal transcribed spacer (ITS)”and the same for line 30 ,,ITS” not ,,ITS-1”. Please check throughout the manuscript
ITS-1 refers to the first of two internal transcribed spacer regions (the first between 18S rRNA and 5.8S rRNA and the second between 5.8S rRNA and 28S rRNA genes). We feel that ITS-1 is more appropriate here as ITS might imply the whole region as opposed to the region between the 18S rRNA gene and the 5.8S rRNA gene that we refer to here.
Line 31
In my opinion should be ,, sequences recovered from strain isolated from Aoudad sheep” and line 32 ,, as well as several sequences recovered from strains isolated from domestic sheep and few other herbivores”
Added the word recovered as suggested by the reviewer. (lines 28, and 30-31 in revised manuscript)
Line 41
,,that collectively mediate in the transformation” sounds better
We respectfully disagree with the reviewer. Mediate is a transitive verb that does not need a preposition.
Line 43
,,Neocallimstigomycota” there is missing letter in this word
Thank you for pointing this out. We have corrected this typo.
Line 90
,,until the next day on which they were used for isolation” is more adequate
Change made as suggested. (line 93 in revised manuscript)
Line 108
I think, there should be ,,obtained isolates”
Thank you for pointing this out. We have corrected this typo.
Line 109, 132
,,as previously described in” – without ,,in”
Thank you for pointing this out. We have corrected this typo.
Line 111-112
Subscripts should be used within molecular formulas of chemical compounds
Thank you for pointing this out. We have corrected these throughout the manuscript.
Line 103 and 115
Please standardize the notation for units, for example Celsius degrees, for comparison: line 103, 115 and 117
Thank you for pointing this out. We have corrected these throughout the manuscript.
Line 121
Should be ,,microscopes”
Thank you for pointing this out. We have corrected this typo.
Line 123, 152
,,an actively growing 2-3 days old culture” sounds better
Thank you for pointing this out. We have changed "2-3 d old" to "2-3 days old" in the two places indicated. (lines 131 and 162 in revised manuscript)
Line 142
There is ,, polygalcturonate” but should be ,,polygalacturonate” also check the line 239
Thank you for pointing this out. We have corrected this typo.
Line 158 (the same in line 185)
There is unnecessary space: 5’- GGAAGTAAAAGTCGTAACAAGG-3` comparing to line 159
Thank you for pointing this out. We have corrected this typo.
Line 219
There is unnecessary space ,,(Figure 2 n)”
Thank you for pointing this out. We have corrected this typo.
Line 273
,,NZ” - all the abbreviations used within the manuscript should be expanded on first use, in this case (I think) the proper place to do this is line 77
We added the abbreviation to the first mention of New Zealand as suggested by the reviewer. (line 80 in revised manuscript)
Line 290
The is ,,phase contract micrographs” but should be ,,phase contrast micrographs”
Thank you for pointing this out. We have corrected this typo.
Line 310, page 10, Figure 4
The name (caption) of clade below Tahromyces is unreadable. The same situation on page 11 below Orpinomyces
Better resolution images are now uploaded. Sorry about that.
Line 313
I think it is worth to add ,,nucleotide sequences”
Done. Line 328 in revised manuscript
Line 373
Please, standardize the notation for units (Celsius degrees)
Thank you for pointing this out. We have corrected these throughout the manuscript.
Line 377
,,tubes were filled to the top with little to no space for air” sound better for me
We changed "room" to "space" as suggested. (line 402 in revised manuscript).